# Monte Carlo simulation of the effect of magnetic fields on brachytherapy dose distributions in lung tissue material

**Fernando Moreno-Barbosa**[1], **Benito de Celis-Alonso**[1], **Eduardo Moreno-Barbosa**[1]*,
**Javier Miguel Hernández-López**[1], **Theodore Geoghegan**[2], **José Ramos-Méndez**[3]

**1** Faculty of Mathematical & Physical Sciences, Benemerita Universidad Autonoma de Puebla, Ciudad Universitaria, Mexico City, Mexico, **2** Department of Radiation Oncology, University of Iowa Hospitals and Clinics, Iowa City, IA, United States of America, **3** Department of Radiation Oncology, University of California San Francisco, San Francisco, CA, United States of America

* emoreno@fcfm.buap.mx

## Abstract

The aim of this work was to use TOPAS Monte Carlo simulations to model the effect of magnetic fields on dose distributions in brachytherapy lung treatments, under ideal and clinical conditions. Idealistic studies were modeled consisting of either a monoenergetic electron source of 432 keV, or a polyenergetic electron source using the spectrum of secondary electrons produced by $^{192}$Ir gamma-ray irradiation. The electron source was positioned in the center of a homogeneous, lung tissue phantom ($\rho$ = 0.26 g/cm$^3$). Conversely, the clinical study was simulated using the VariSource VS2000 $^{192}$Ir source in a patient with a lung tumor. Three contoured volumes were considered: the tumor, the planning tumor volume (PTV), and the lung. In all studies, dose distributions were calculated in the presence or absence of a constant magnetic field of 3T. Also, TG-43 parameters were calculated for the VariSource and compared with published data from EGS-brachy (EGSnrc) and PENELOPE. The magnetic field affected the dose distributions in the idealistic studies. For the monoenergetic and poly-energetic studies, the radial distance of the 10% iso-dose line was reduced in the presence of the magnetic field by 64.9% and 24.6%, respectively. For the clinical study, the magnetic field caused differences of 10% on average in the patient dose distributions. Nevertheless, differences in dose-volume histograms were below 2%. Finally, for TG-43 parameters, the dose-rate constant from TOPAS differed by 0.09% ± 0.33% and 0.18% ± 0.33% with respect to EGS-brachy and PENELOPE, respectively. The geometry and anisotropy functions differed within 1.2% ± 1.1%, and within 0.0% ± 0.3%, respectively. The Lorentz forces inside a 3T magnetic resonance machine during $^{192}$Ir brachytherapy treatment of the lung are not large enough to affect the tumor dose distributions significantly, as expected. Nevertheless, large local differences were found in the lung tissue. Applications of this effect are therefore limited by the fact that meaningful differences appeared only in regions containing air, which is not abundant inside the human.

**Data Availability Statement:** The data underlying the study's findings are available from a third party source; Data from iridum source are available from

https://physics.carleton.ca/clrp/seed_database/Ir192_HDR/VariSource_VS2000.

**Funding:** Authors would like to thank CONACyT for the support of Ph.D. student FMB. JRM was partially funded by UCSF School of Medicine bridge fundingand National Cancer Institute under U24 CA215123 (The TOPAS Tool for Particle Simulation, a Monte Carlo Simulation Tool for Physics, Biology and Clinical Research). The authors thankfully acknowledge computer resources, technical advice and support provided by the Laboratorio Nacional de Supercómputo del Sureste de México (LNS), a member of the CONACYT national laboratories. The funders had no role in study design, data collection and analysis, decision to publish, or preparation of the manuscript.

**Competing interests:** The authors have declared that no competing interests exist.

# Introduction

In 1895, H.A. Lorentz mathematically described the theory that describes the effect that magnetic and electric fields have on moving charged particles. This electromagnetic theory states that the trajectory of a charged particle is curved perpendicularly to the plane containing the velocity and magnetic field vectors. In medical physics, this effect is relevant because of two facts. First, the presence of strong magnetic fields in clinical environments. Clinical scanners based on magnetic resonance (MR) use fields between 1.5 T and 3 T, while higher fields up to 11.7 T or more can also be found in some specialized research environments. Second, the use of image-guided radiotherapy and brachytherapy treatments as part of the clinical workflow. The enhanced contrast/resolution provided by MR imaging is an upgrade to those of traditional use like X-ray imaging. Therefore, the industry is already promoting the production of combined MRI-Linac machines. In this context, there are valid and interesting questions to be considered. For example, what are the effects that magnetic fields induce on dose distributions produced by charged particles? Is it safe for patients and health professionals to work in this environment? Can the Lorentz force be used to limit dose distributions and confine them to smaller and more precise volumes? Is the dose distribution homogeneous inside this confined area? Computational modeling using analytical or Monte Carlo (MC) methods may assist in addressing such questions through the precise control of the theoretical variables of the physics processes and geometry models involved. For example, in-patient dose distributions in regions-of-interest with useful resolution can only be obtained with computational tools (~1 mm$^3$ voxels size).

The effects of magnetic fields on electron beams produced by clinical linear accelerators (LINAC) and their interaction with patients have been analyzed with MC methods in the past [1–3]. For LINACs, where the particles were emitted in one preferential direction, findings showed reduced scattering due to the helicoidal movement of electrons along the direction of the magnetic field (**B**). In some cases, it was possible to optimize the dose distributions using this effect. When the direction of the beam was transversal to the magnetic field, this effect was found to be larger [4, 5]. Differences between dose distributions of up to 300% were found under the presence of magnetic fields ranging from 1.5 T to 11 T. When dose distributions were studied inside an MRI scanner, two effects became apparent to researchers. The first effect was the contribution of secondary electrons leaving and then returning to the tissue caused by the bending of their trajectories [4, 6, 7]. The second was the shorter dose build-up regions needed due to the bending of electron trajectories [5, 6]. All these effects varied linearly with the magnetic field strength and the particle energies involved, and also depended on the tissue composition. In a more recent study, Beld et al [8] showed the effects of magnetic fields on dose distributions produced by a $^{192}$Ir high-dose rate (HDR) brachytherapy source. Differences from previous studies arose from the fact that the brachytherapy source emitted radiation isotopically, and the energy spectrum produced by the source was lower than that used in LINACs (keV vs. MeV). Results were obtained in air and water, representing the rectum cavity and prostate tissue, respectively. They reported that the clinical effect in water was negligible, but relevant changes in dose distributions near the air pockets were found. This study covered two extreme cases where two homogeneous materials were used with density values three orders of magnitude apart.

To our knowledge, the case of intermediate, low-density tissue has not been reported. A particularly important case is the lung, as it is composed of a heterogeneous distribution of soft and dense tissues ranging from air to compressed lung [9]. Given the existence of that density distribution, physical changes in the dose distributions are expected according to the extreme cases presented in Beld et al, and call for their quantification.

The aim of this work was to use MC simulations to assess the effect on dose distributions during the course of $^{192}$Ir brachytherapy treatments of lung inside an MR scanner. Its application specifically to lung cancer is justified on two facts. First, the prevalence of this type of cancer and its mortality which accounts for 14% of all cancers diagnosed and 25% of all annual, cancer-related deaths [10–12]. Second, the larger "confining" effects that are expected to be found in low-density tissue and in the air environment, due to Lorentz forces. To this end, dose distributions were calculated in a homogeneous low-density lung material using mono- and poly-energetic electron spectra. This allowed evaluation of the best- and worst-case scenarios in the presence or absence of an external uniform magnetic field. Subsequently, an array of HDR brachytherapy sources of $^{192}$Ir were modeled for a clinical lung treatment under the presence of a homogeneous magnetic field. This allowed heterogeneous tissue densities and their spatial distribution to be accounted for. All the modeling was performed with the Geant4-based TOPAS MC tool [13, 14]. This allowed the implementation of detailed MC simulations considering the physics of particle interactions with matter, electromagnetic fields and patient geometries through a user-friendly interface.

## Methods

### Monte Carlo software

The modeling software used was TOPAS [14] version 3.2, built on top of the Geant4 MC toolkit [13] version 10.5 patch 1. The radioactive decay process of iridium was modeled using the Geant4 G4RadioactiveDecay constructor. This physics constructor contains verified physical models to simulate radioactive decay of many radionuclides, as reported in Hauf et al [15]. The electromagnetic interactions of photons, electrons, and positrons were modeled using Geant4's constructor G4EmStandard_option4. For photons, this physics constructor used Livermore models for Rayleigh scattering and the photoelectric effect, a low energy model for Compton scattering [16], and models based on the PENELOPE MC code [17, 18] for gamma conversion. For electrons and positrons, the constructor used PENELOPE models for the ionization process below 1 MeV, and the Goudsmit-Saunderson model for multiple scattering. For the transport of charged particles in magnetic fields, the Geant4 parameters consisted of the classical four-order Runge-Kutta (ClassicalRK4) stepper with DeltaChord value of 0.05 mm. Additional parameters were dRoverRange and finalRange with values of 0.2 and 10 μm, respectively. Experimental comparison has been performed showing reasonable agreement for electrons produced by a clinical LINAC using a 0.1 mm DeltaChord value and the ClassicalRK4 stepper [7]. All the simulations were run in the Mexican Southeast National Laboratory of Supercomputing (LNS). A computational setup consisting of 228 nodes with 5472 CPU cores.

### Comparison of TOPAS with EGS-brachy and Penelope for an $^{192}$Ir seed scenario

TOPAS, from its initial stages, has been developed to be used in proton therapy applications [14]. Nevertheless, the physical models and geometrical tools provided by this tool allow its use for other radiotherapy modalities, including brachytherapy. Therefore, TOPAS was compared to Monte Carlo modeling standards in brachytherapy, making the results of this work more reliable. To this end, the authors compared calculated dose distributions and AAPM TG-43 [19] parameters with published data calculated with PENELOPE and EGS-brachy in similar conditions [20, 21]. The TG-43 parameters compared were dose-rate value in water, air-kerma strength, and the geometry and anisotropy functions. For dose distribution

calculations, a cylindrical water phantom with a 15 cm radius and 15 cm height was used. The phantom was divided into bins with 0.05 cm resolution in both the radial and longitudinal axes. This resolution was the same as the one used in PENELOPE calculations [20]. For air-kerma strength calculations, the fluence spectrum was scored *in-vacuo* on the surface of a binned sphere of 100 cm radius. The polar angle was covered with 1-degree resolution differentials. The fluence spectrum was then weighted with the air mass energy-absorption coefficients obtained from NIST [22], according to the updated AAPM TG-43 report [19]. A low energy limit of 5 keV was used.

## The effect of a magnetic field on dose distributions of [192]Ir in homogeneous lung tissue

As a first step, the authors considered two idealistic scenarios, each using an electron source interacting in a cubic phantom made of homogeneous lung tissue. For the first scenario, a punctual, isotropic and monoenergetic electron source of 432 keV was positioned in the center of a cubic lung tissue phantom of homogeneous density. A constant lung density of 0.26 g cm$^{-3}$ was calculated as an average from its deflated and inflated states [9]. By using this value, the necessity of making simulations for either inflated (mean density of 0.1 g cm$^{-3}$) and deflated (mean density of 0.4 g cm$^{-3}$) lung tissue was eliminated. The atomic composition of the lung was obtained from the International Commission for Radiation Protection database [23]. This information was available in TOPAS through the G4_LUNG_ICRP nomenclature of Geant4 [13]. The energy value of 432 keV corresponded to the maximum energy transferred to secondary electrons set in motion by [192]Ir gamma-rays interacting in water [8]. For the second scenario, the energy of the electrons was replaced by the whole energy spectrum obtained from Fig 1A in Beld et al. [8]. Thus, the first scenario allowed the authors to evaluate the largest effect on the dose distributions. The effect was largest due to the longer range that electrons could reach given the maximum energy of the spectrum. Nevertheless, this was not the most likely scenario, and therefore, the second scenario allowed the authors to evaluate the effect on the dose distributions using the most probable range of electron energies.

For both scenarios, the simulated phantom was a 1 cm$^3$, voxelized cubic box with a 0.05 mm x 0.05 mm x 0.05 mm resolution. The radioactive source was positioned in the center of the phantom, and two magnetic strengths were applied: 0 T, or no magnetic field, and 3 T magnetic field. For all scenarios, the magnetic field was pointed in the positive **z**-direction. A production cut off value of 0.05 mm was used for the creation of secondary electrons, which is the default value in TOPAS.

Three-dimensional dose distributions were calculated and normalized to a dose value averaged over an arbitrary point, located at coordinates (x = 1 mm, y = z = 0 mm). Results were presented as normalized dose distributions with green lines being the 100% iso-dose, followed by 50%, 25%, 10%, and 1% iso-dose lines (in blue tones). In total, 200 million histories were simulated to achieve an averaged statistical uncertainty lower than 0.5% at 100% of the normalized dose distributions.

## Clinical brachytherapy treatment plan using [192]Ir

A [192]Ir brachytherapy seed was modeled for this work. The brachytherapy seed reproduced the commercially available iridium seed VariSource VS2000 HDR from Varian®. This seed is regularly used with a VariSource™ iX Afterloader from Varian® for brachytherapy treatments [24]. The geometry details were obtained from a previous publication [20]. The external dimensions of the seed were 6.59 mm long and 0.59 mm wide, with the iridium core encapsulated inside (see Fig 1A). The core consisted of two cylinders of 0.17 mm radius and 2.5 mm

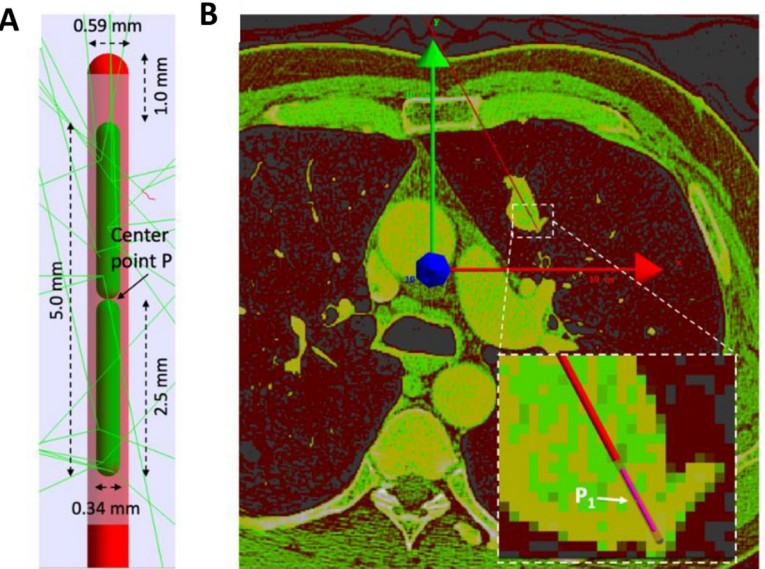

**Fig 1. Geometric characteristics of VS2000 source & setup array in lung tumor.** Panel A. TOPAS simulation of the brachytherapy seed VS2000 immersed in water. The casing is depicted with red and light red, and the [192]Ir radionuclide source in green. Real dimensions of the seed were used for this simulation. A few gamma-rays and electron tracks are shown with green and red lines, respectively. The center point of the seed (P) is shown with a black arrow. Panel B. TOPAS simulation of a lung patient calculated from a DICOM CT image. Brachytherapy seed centered at position $P_1$ within the tumor. The global coordinate system is overlaid on the modeled image. The red arrow points towards the x-axis, the green arrow towards the y-axis and the blue arrow to the z-axis. In the zoomed area, the seed positioning cable is represented in red and the seed in pink.

length, including rounded endcaps with a 0.295 mm radius. The shielding material of the source was a titanium-nickel alloy (44:56 fraction by weight), with a density of 6.5 g cm$^{-3}$. The iridium density inside the shielding was 22.42 g cm$^{-3}$.

A clinical case of pulmonary cancer treated with [192]Ir brachytherapy sources was simulated (see Fig 1B). First, anatomical images in DICOM format of a pulmonary tumor were obtained from the image bank of the Cancer Imaging Archive [25]. The selected DICOM images corresponded to an axial set of computed tomography (CT) slices crossing a pulmonary tumor (voxel resolution: 0.69 mm x 0.69 mm x 1.00 mm, 60 slices). One advantage of using this data was that the tumor was confirmed by the research group in charge of uploading the images, and they provided information on node location and diagnosis. Hounsfield units from the DICOM files were converted to density values using the Schneider stoichiometric method [26]. This procedure is automated in TOPAS [27]. The simulation moved away from the constant lung density of 0.26 g cm$^{-3}$ used in previous models, to a more realistic scenario with density values ranging from 0.01 g cm$^{-3}$ to 0.6 g cm$^{-3}$.

For HDR interstitial brachytherapy of lung lesions, the use of a single brachytherapy catheter in tumors up to a 4 cm diameter, has been previously reported [28, 29]. The use of a single catheter showed a 0% complication rate of pneumothorax and a 75% tumor control rate for 20 Gy of prescribed dose in a single fraction [29]. Thus, in this work, four seed positions in a single catheter were used to deliver a 20 Gy single dose to the planning tumor volume (PTV). On the other hand, automatic brachytherapy planning can be optimized using multicriteria optimization algorithms that demand high-speed computing capabilities, see Belanger et al. [30]. However, it is still a regular practice in brachytherapy planning to manually fine-tune the seed positions and dwells times to obtain a case-specific valid plan [31]. In this sense, the catheter

**Table 1. Seed positions and weights (see the text) used for HDR lung treatment of clinical case.**

| POSITION | COORDINATES (X, Y) CM | WEIGHT (RESPECT TO SOURCE EMISSION) |
|---|---|---|
| $P_1$ | (4.423, 2.012) | 1.370 |
| $P_2$ | (4.095, 2.630) | 2.518 |
| $P_3$ | (3.766, 3.248) | 1.296 |
| $P_4$ | (3.437, 3.866) | 1.000 |

The coordinates are given with respect to the global coordinate system. Each coordinate corresponds to the center of the seed located between the two cylinders that c composed the core, (point P1 in Fig 1B). For all positions, the seed made a negative angle of 152 degrees with respect to the y-axis. The position in coordinate z was -0.141 cm (approx. middle of the CT DICOM image stack). Values in the third column are weights with respect to source emission. Values in the fourth present % weight of total number of histories of each source.

angle and the seed positions were selected to cover the 2.8 cm diameter tumor volume. For the dwell times, each seed position was maintained for a unit of time (weight = 1) or multiple units of time (weight > 1) from a position of reference in order to obtain a tumor dose distribution that was as uniform as possible. This was accomplished by weighting the number of histories at each source position. The seed positions and weights are presented in Table 1.

The transport of radiation within the seed was handled by the parallel navigation of Geant4 and facilitated by TOPAS. The seed material was designed according to the layered mass geometry feature [32]. In this way, the simulation setup was simplified as there was no need to use voxels filled with water to accommodate the seeds as reported previously [33].

Since the millions of decays modeled in MC methods correspond to small amounts of time, the activity of the sources was assumed to be constant during the simulation period [34]. The simulations were performed with and without a uniform 3 T magnetic field pointing in the caudal-cephalic direction (positive **z**). In total, $3 \times 10^{10}$ histories distributed in 30 jobs were simulated for each option (0 T and 3 T fields). That number of histories was enough to obtain dose distributions with statistical uncertainties better than 0.5% and 1%, for voxels with a dose bigger than 50% or 10% of the prescribed dose, respectively. The dose distributions were normalized to the prescribed 20 Gy dose at the PTV structure. Three structures of interest were contoured: the tumor, the planning target volume (PTV), and the lung. The volumes of these structures were 3.9 cm$^3$ for the tumor, 12.8 cm$^3$ for the PTV, and 895.2 cm$^3$ for the lung (from a DICOM selected number of slices). Finally, dose-volume histogram parameters suggested for interstitial brachytherapy were calculated [35]. $V_{90}$, $V_{100}$, $V_{150}$, $V_{200}$, $D_{90}$, and $D_{100}$ were calculated for the tumor and PTV structures, and $D_{2cc}$ for the lung. Anatomical images underneath the dose images were co-registered to the same space and presented at the same resolution.

## Statistics

All results are presented as a mean ± SEM (standard error of the mean). While comparison between populations was necessary, data distributions were first assessed for normality, and depending on the result either Student's t-tests or Mann-Whitney U tests were applied. Significance of the results was considered to have been achieved when $p < 0.05$. For all calculations, SPSS 9 software (IBM®) was used.

## Results

Fig 2 shows the modeling results of a 432 keV electron source emitting isotopically in lung tissue. As depicted in the top panels of the image, in the absence of a magnetic field, the dose distribution decreased radially as the inverse of the squared distance. This decreasing in dose is

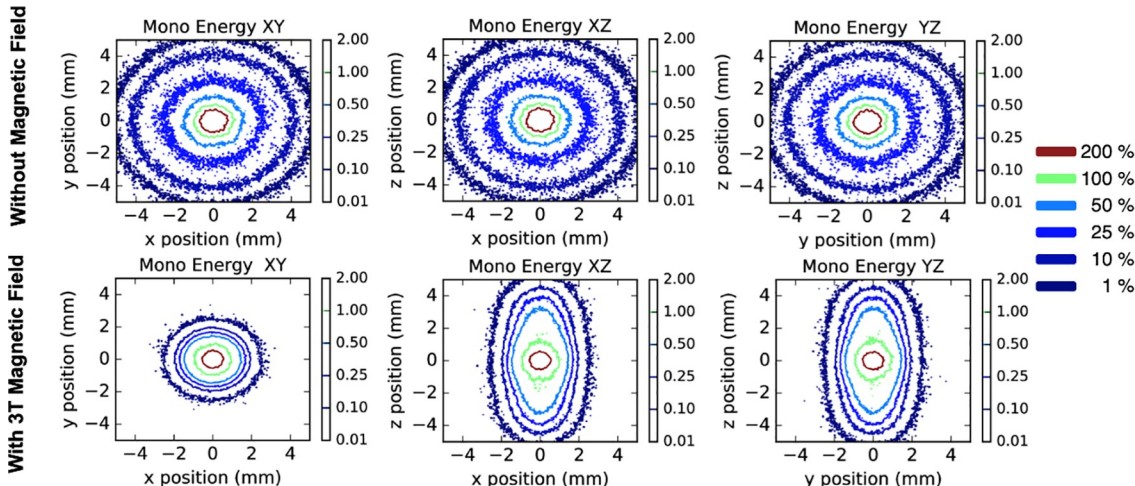

**Fig 2. Monoenergetic electron source (432 keV) in a homogeneous lung environment.** Normalized dose distributions in a homogeneous lung-tissue phantom for a monoenergetic electron source. Results are shown in absence of a magnetic field (top row), and under the presence of a 3 T magnetic field (bottom row) oriented on the positive **z**-axis. Simulations are limited to ± 5 mm regions in their corresponding planes.

represented by a change in color from red to dark blue. When a magnetic field was applied in the positive **z**-direction (bottom panels), electrons moving along the +**x**-direction were subject to a force in the -**y**-direction, and electrons moving in the +**y**-direction were subject to a force in the +**x**-direction, due to the Lorentz force. In this way, there was a helicoidal movement in the XY plane. This movement caused the iso-doses to be distributed in both directions (x and y) from 1.02 mm ± 0.06 mm to 0.94 mm ± 0.05 mm distance for the 100% iso-dose, and from 4.08 mm ± 0.26 mm to 1.43 mm ± 0.05 mm for the 10% iso-dose. These differences were statistically significant for both cases (p<0.001) and represented 7.8% and 64.9% reductions, respectively. The helicoidal movement in the XY plane caused the pattern in the iso-dose distribution shown in the XZ and YZ panels.

The results of modeling the spectrum of secondary electrons from $^{192}$Ir gamma-rays in pulmonary tissue can be seen in Fig 3. As seen in the top panels of Fig 3, with the absence of a magnetic field, the dose distribution was isotropic, decreasing as the inverse of the square root of distance. Because the electron energies of this spectrum were a composite of energies in the 0–432 keV range, the dose deposition was closer to the source position than in the scenario using the monoenergetic electron source. When a magnetic field was applied in the positive **z**-direction (bottom panels), the same effect appeared (Fig 2 bottom panels) even though it was of a smaller magnitude. In the cases without magnetic field, the iso-doses at 100% and 10% were distributed at 1.00 mm ± 0.06 mm and 2.01 mm ± 0.29 mm, respectively. In the cases with the magnetic field, the iso-doses at 100% and 10% were redistributed to 0.92 mm ± 0.02 mm and 1.48 mm ± 0.10 mm in both directions (X and Y), respectively. These differences were statistically significant for both cases (p<0.001) and represented 8.0% and 26.4% reductions, respectively. The differences in the XZ and YZ planes due to the magnetic field were of the same magnitude in either x or y direction and are not presented for clarity.

In Fig 4, the TG-43 parameters shown are the geometry function ($G_L$), anisotropy function (F), and dose-rate constant (Λ). For the geometry and anisotropy functions, the point-to-point ratios of TOPAS to EGS-brachy published calculations are displayed at the bottom of panels A and B, respectively. The geometry function calculated with TOPAS and EGS-brachy differed by 1.2% ± 1.1% at 15 cm, one standard deviation. For the anisotropy function, results from

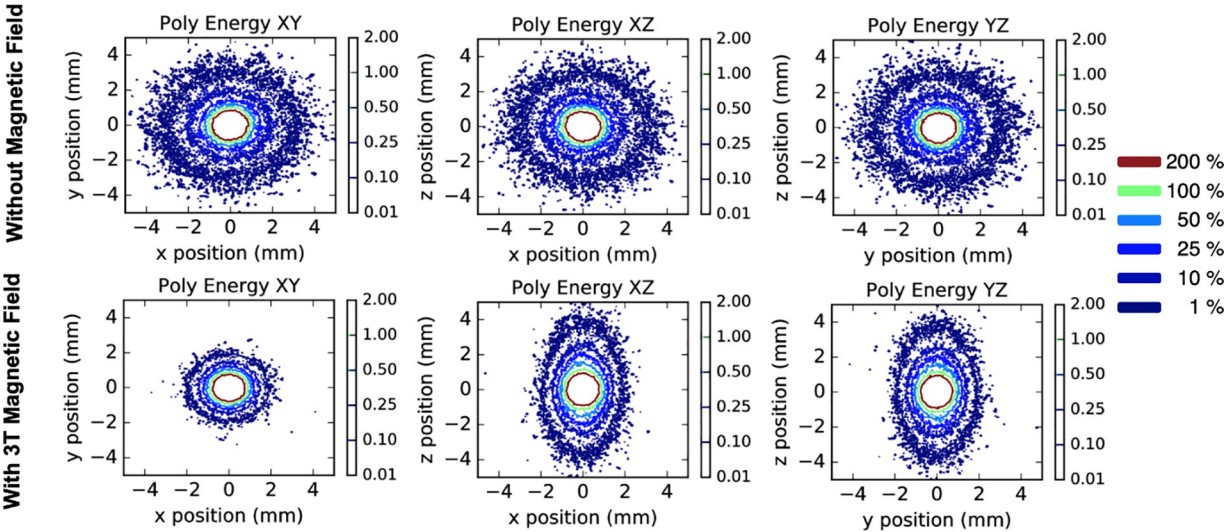

**Fig 3. Poly-energetic electron source in a homogeneous lung environment.** Normalized dose distributions in a homogeneous lung-tissue phantom for a poly-energetic electron source. Results are shown in absence of a magnetic field (top row), and under the presence of a 3 T magnetic field (bottom row) oriented on the positive **z**-axis. Simulations are limited to ± 5 mm regions in their corresponding planes.

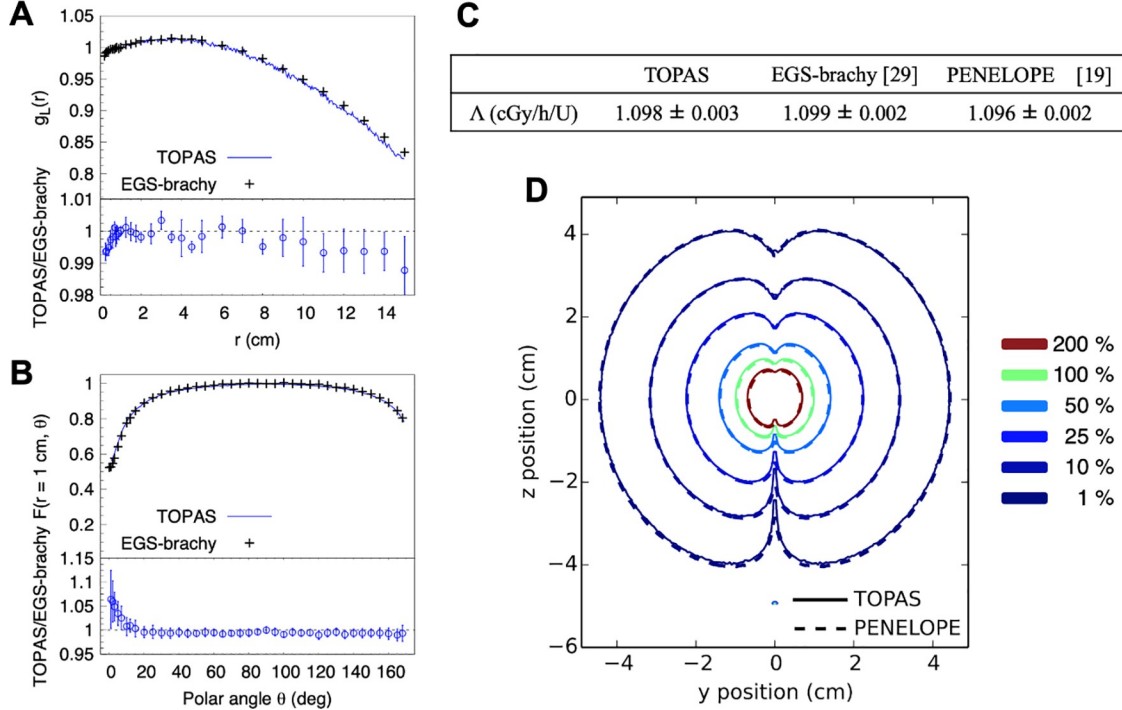

| | TOPAS | EGS-brachy [29] | PENELOPE [19] |
|---|---|---|---|
| Λ (cGy/h/U) | 1.098 ± 0.003 | 1.099 ± 0.002 | 1.096 ± 0.002 |

**Fig 4. Comparison TOPAS vs. EGS-brachy & Penelope.** TG-43 parameters are shown in panels A-C. The geometry function (panel A) and anisotropy function (panel B) are shown with point-to-point ratios, with TOPAS to EGS-brachy, at the bottom of each panel. Displayed errors bars and errors are statistical uncertainties of one standard deviation. In panel C, the dose-rate is shown and compared to EGS-brachy [21] and PENELOPE [20]. Panel D. presents dose distributions normalized at y = 1 cm, z = 0. Continuous iso-dose lines correspond to TOPAS while dashed lines correspond to PENELOPE calculations from Almansa et al [20].

both codes were within statistical uncertainties, except for angles smaller than 6 degrees where TOPAS exceed EGS-brachy calculations by 6.4% ± 6.1% at 0 degrees. For the dose-rate constant, differences with respect to EGS-brachy and PENELOPE calculations were 0.09% ± 0.33% and 0.18% ± 0.33%, respectively. Finally, panel D shows the dose distributions on the central plane of the brachytherapy seed. The good agreement between iso-doses and the effect of the wire attached to the source in the lower region can be visually appreciated.

Fig 5 shows the brachytherapy treatment case modeled with an $^{192}$Ir source using the configurations from Table 1. Fig 5A shows the iso-dose curves superimposed on the anatomical CT image with dose values displayed in percent of the prescribed dose. Note that the 100% iso-dose line covered the tumor completely. The volume of lung tissue receiving 5 Gy was 19.5%, a volume consistent with the clinical use case reported in Sharma et al. [28]. As shown, the iso-dose curves of simulations with and without a 3 T magnetic field, overlap in denser tissues but are misaligned in softer tissues. To quantify this difference, Fig 5B shows the relative dose differences in each voxel with respect to the simulation without the magnetic field. Only those voxels with differences bigger than 2%, and one standard deviation of combined statistical uncertainty are displayed. One average, differences *circa* 10% are visible at the distal area of seed position $P_1$ (Fig 1B) with a hot spot of more than 25%. At the boundaries between lung and muscle tissue, differences of 5% on average can be found. DVH´s obtained for the tumor, PTV, and lung are shown in Fig 6. The DVH´s and point-to-point differences between no field and 3 T field results are also shown. Systematic differences outside 1 standard deviation were found for the tumor below 1% (dose > 35 Gy), the PTV below 1.5% (dose > 45 Gy), and the lung below 2% (dose > 40 Gy).

The dose-volume histogram parameter $V_{100}$ for the tumor with and without the magnetic field was 100% in both cases. For the PTV, $V_{100}$ was 97.32% ± 0.09% (without field) and 97.20% ± 0.07% (with field). Finally, for the lung, the parameter $D_{2cc}$ was 96.79% ± 0.11% and 97.95% ± 0.14% (1.18% ± 0.12% difference). Differences in percentage for all the parameters are summarized in Table 2. As shown, except for $D_{100}$, differences were below 1% (one standard deviation) for the parameters found for the tumor and PTV. For parameter $D_{100}$, the

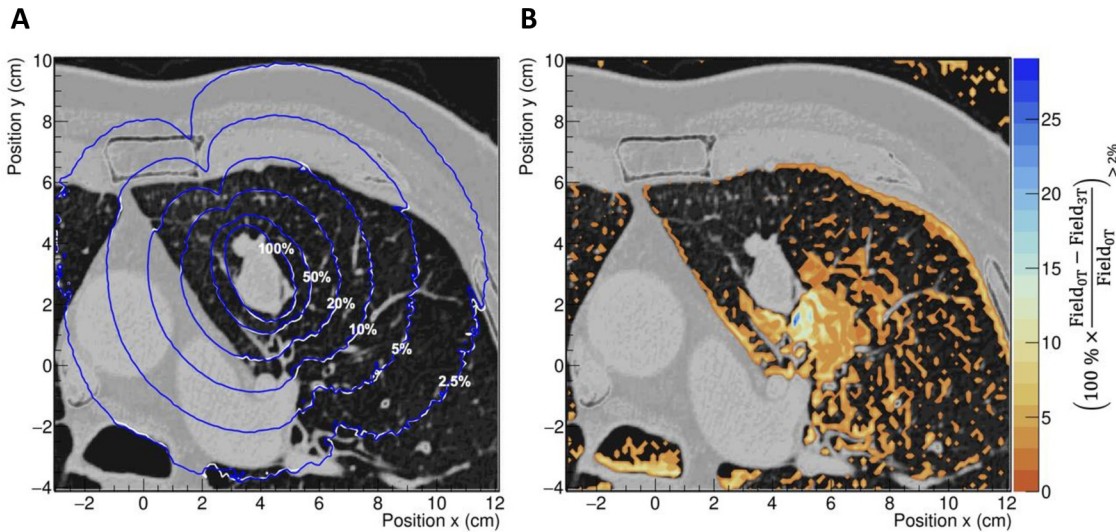

**Fig 5. The effects on dose distribution of a magnetic field in a clinical case of lung tumor. A**: dose distribution for the simulations in presence of a 3T magnetic field (blue lines), and in absence of magnetic field (white lines). Iso-dose curves are displayed in percentage of the prescribed dose. **B**: relative differences with respect to the no magnetic field simulation. Only relative differences larger than the combined statistical uncertainty of one standard deviation, are depicted.

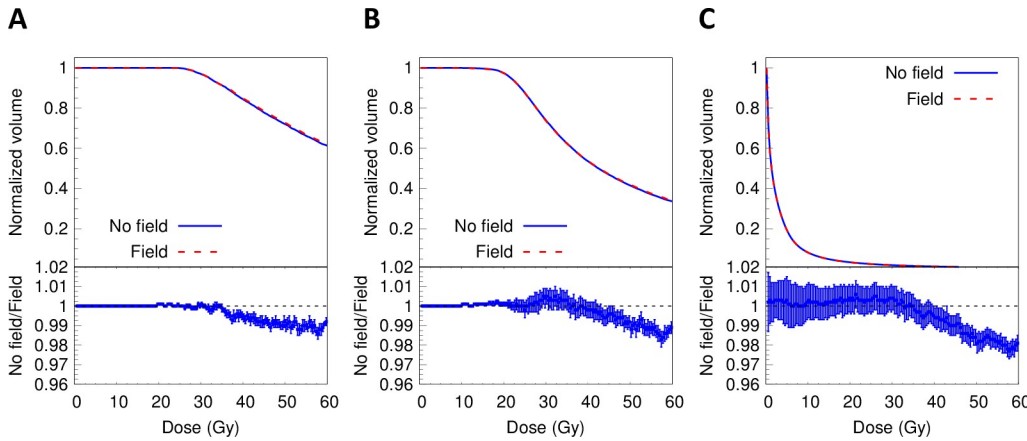

**Fig 6.** Dose-volume histograms (DVH) for the tumor volume (A), PTV volume (B) and lung volume (C). Point-to-point ratios between results with a 3T field to results with no field are shown at the bottom of each figure. Error bars represent statistical uncertainties, one standard deviation.

presence of the field reduced the dose by 11.43% and 13.95% for the Tumor and PTV, respectively. These results show that clinically, inside the tumor, there were no major differences in dose distributions due to the presence of a magnetic field.

## Discussion

This work presented a theoretical assessment using Monte Carlo simulations of the effect of magnetic fields on dose distributions from brachytherapy. The Monte Carlo method has proved its validity when an experimental measurement is difficult to perform, as in the assessment of theoretical predictions, development of prototyped technology, and analysis of uncertainties [6, 8, 36, 37]. In these kinds of applications, well-controlled conditions and prior verification of the components involved (physics models and geometry models) must be ensured. In this work, such conditions were satisfied in the following way. First, the TOPAS Monte Carlo model of the brachytherapy source was verified by comparing calculated TG43 parameters with data from the literature. Second, this work relies on the accuracy of Geant4 toolkit to model transport of charged particles in magnetic fields, which has been already shown experimentally [7, 38, 39] and theoretically [40, 41]. Third, we relied on the TOPAS tool for the implementation of magnetic field, DICOM patient, detailed geometry and source of the brachytherapy seed, and scoring. TOPAS minimizes the possible error introduced by the user by providing a robust user interface [14]. Finally, the simulated scenarios were performed in well-controlled conditions by varying only the magnetic field. Authors acknowledge that the lack of a real physical experiment supporting the findings presented here, even if properly argued and justified, will always be a limitation of any kind of study which uses modeling.

**Table 2. Percentage differences between DVH parameters from simulations with and without magnetic field.**

|  | $\Delta V_{90}$ (%) | $\Delta V_{100}$ (%) | $\Delta V_{150}$ (%) | $\Delta V_{200}$ (%) | $\Delta D_{90}$ (%) | $\Delta D_{100}$ (%) |
|---|---|---|---|---|---|---|
| **Tumor** | 0.00 ± 0.01 | 0.06 ± 0.01 | 0.06 ± 0.11 | -0.57 ± 0.22 | -0.52 ± 0.35 | 11.43 ± 0.95 |
| **PTV** | 0.16 ± 0.04 | 0.12 ± 0.11 | 0.43 ± 0.45 | -0.21 ± 0.41 | 0.04 ± 0.48 | 13.95 ± 2.33 |
| **Lung** | 0.17 ± 1.12 | 0.20 ± 1.13 | 0.24 ± 1.17 | -0.64 ± 1.33 | 0.10 ± 0.38 | 0.00 ± 0.01 |

Errors are statistical, 1 standard deviation.

Based on the conditions described before, the main findings of this work were as follows. The dose distributions from an [192]Ir source irradiating lung tissue with a homogeneous density were affected by the magnetic field. The effect occurred using either monoenergetic 432 keV electrons or the polyenergetic spectrum. This effect reduced the distance, from the source initial position, of the 10% iso-dose line by 64.9% for monoenergetic electrons and 26.4% for the whole spectrum. These differences motivated the exploration of a clinical case, where a heterogeneous density distribution existed. In this case, the difference in dose-volume histograms of the tumor with and without magnetic fields was small. However, for PTV and lung regions which received high doses of radiation (100–300% of planned), differences in dose-volume histograms of 0% to 3% were found. Finally, the TOPAS tool agreed with results from EGS-brachy and PENELOPE codes when modeling the dose depositions of a brachytherapy seed.

The magnetic field locally affected the energy deposition in the lung tissue. In Fig 5B, a hot region close to the tumor volume is shown. The intensity was accentuated in regions closer to the tumor. In lung tissue, the produced electrons had extended ranges compared to electrons of the same energy produced in denser tissue. Hence, the bending effect of Lorentz forces on the trajectory of electrons in soft tissue was more pronounced. This caused a more confined dose distribution compared to the distribution calculated in the absence of the magnetic field. At the same spatial distance far from the source, the dose was higher in the absence of a magnetic field, as shown in Fig 3. This was also present in the ratios of dose-volume histograms shown in Fig 6. The electrons produced in the denser tissue that reach the soft tissue experienced the electron return effect [8] and produced hot spots near the interfaces, as seen in Fig 5B.

The tumor volume did not show major differences in dose distributions at almost any prescribed dose due to the presence or absence of the magnetic fields, as shown in Fig 6. This was an expected result caused by the tumor density. The tumor, because of its vascularity, presents larger amounts of blood (fluid) rather than air. Beld at al. [8] already demonstrated that the bending effect in water was negligible. Our results were consistent with Beld's findings.

TOPAS was suitable for brachytherapy calculations in homogeneous water. Although TOPAS was developed for proton therapy applications, its features and Geant4 physics models allowed its use in other radiotherapy modalities. In this work, we verified the feasibility of using TOPAS through the calculation and comparison of the TG-43 parameters. Comparison of TOPAS calculations was performed with results from EGS-brachy and PENELOPE codes for the VariSource VS2000 [192]Ir seed. For the geometry function, systematic differences of 0.62% with respect to EGS-brachy were found at positions closer to the seed ($r < 0.5$ cm). These were caused by the voxel size used in this work (0.5 mm) compared to that used in EGS-brachy (0.1 mm $< 1$ cm). The systematic error introduced by the voxel size was investigated previously in [42]. In there, differences of 1.8% at 0.3 cm were found when going from scoring spherical shells of thickness 0.1 mm to 0.5 mm. The differences found in this work were consistent with the reported error from [42]. The geometry size effect was also responsible for the highest differences found in the anisotropy function at angles closer to 0 degrees. Nevertheless, the dose-rate calculated in this work agreed with other MC codes within statistical uncertainties of 0.33%.

## Conclusions

Differences between models with and without magnetic field were minimal (absolute difference of approximately 3.0% as it can be derived from Fig 6C). Based on these results it can be argued that the concomitant use of a magnetic field in brachytherapy did not bring any advantage to this specific clinical lung case. Nevertheless, it is important to highlight the existence of

a major redistribution of dose deposition in other tissues due to the magnetic fields. At the moment, because the Lorentz effect is only significant for air cavities, very high energy deliveries or strong magnetic fields, there are limits to its applicability in clinical brachytherapy.

## Acknowledgments

The authors thankfully acknowledge computer resources, technical advice and support provided by the Laboratorio Nacional de Supercómputo del Sureste de México (LNS), a member of the CONACYT national laboratories.

## Author Contributions

**Conceptualization:** José Ramos-Méndez.

**Data curation:** Javier Miguel Hernández-López.

**Formal analysis:** Benito de Celis-Alonso.

**Investigation:** Eduardo Moreno-Barbosa.

**Methodology:** Fernando Moreno-Barbosa.

**Software:** Fernando Moreno-Barbosa.

**Supervision:** José Ramos-Méndez.

**Validation:** José Ramos-Méndez.

**Writing – original draft:** Fernando Moreno-Barbosa, Benito de Celis-Alonso.

**Writing – review & editing:** Eduardo Moreno-Barbosa, Javier Miguel Hernández-López, Theodore Geoghegan.

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
