## [Decision Letter · Decision Letter 0]

24 Aug 2020

Monte Carlo simulation of the effect of magnetic fields on brachytherapy dose distributions in lung tissue material

PONE-D-20-21759

Dear Dr. Moreno Barbosa,

We’re pleased to inform you that your manuscript has been judged scientifically suitable for publication and will be formally accepted for publication once it meets all outstanding technical requirements.

Kind regards,

Rongxiao Zhang

Academic Editor

PLOS ONE

Additional Editor Comments (optional):

I agree with both reviewers that the paper is well-written on an interesting topic.

Reviewers' comments:

Reviewer's Responses to Questions

**Comments to the Author**

1. Is the manuscript technically sound, and do the data support the conclusions?

Reviewer #1: Yes

Reviewer #2: Yes

2. Has the statistical analysis been performed appropriately and rigorously? 

Reviewer #1: Yes

Reviewer #2: Yes

3. Have the authors made all data underlying the findings in their manuscript fully available?

Reviewer #1: Yes

Reviewer #2: Yes

4. Is the manuscript presented in an intelligible fashion and written in standard English?

Reviewer #1: Yes

Reviewer #2: Yes

5. Review Comments to the Author

Reviewer #1: Excellently written paper about a well-done study. Although the findings are negative (magnetic field does not impact the dose distribution significantly for Ir-192 brachytherapy sources) it's an important finding to establish as MRI becomes more prevalent in the clinic. I might have selected a nasopharynx cancer rather than a lung tumor treatment for the clinical example as more clinically relevant. Is the lung treatment approach shown in Figure 1B actually clinical? It appears the catheter would have to go through the sternum. Overall I found the study to be complete; the authors address the one big limitation in the discussion, that there are no physical measurements to compare against.

Reviewer #2: This is a thorough and well written paper that carefully considers the effects of magnetic field on the dose deposition of the Ir-192 source in lung tissue. The text and figures provided are clear, with appropriate detail. Appropriate background information, references, description and validation of methods, results, and conclusions are provided. No necessary revisions are identified.

6. PLOS authors have the option to publish the peer review history of their article (what does this mean?). If published, this will include your full peer review and any attached files.

Reviewer #1: No

Reviewer #2: No

---

## [Editor Report · Acceptance letter]

10 Sep 2020

PONE-D-20-21759 

Monte Carlo simulation of the effect of magnetic fields on brachytherapy dose distributions in lung tissue material 

Dear Dr. Moreno-Barbosa:

I'm pleased to inform you that your manuscript has been deemed suitable for publication in PLOS ONE. Congratulations! Your manuscript is now with our production department. 

Kind regards, 

on behalf of

Dr. Rongxiao Zhang 

Academic Editor

PLOS ONE